# In Vivo Anti-Inflammatory Effects and Related Mechanisms of Processed Egg Yolk, a Potential Anti-Inflammaging Dietary Supplement

**DOI:** 10.3390/nu12092699

**Published:** 2020-09-04

**Authors:** Joan Cunill, Clara Babot, Liliana Santos, José C. E. Serrano, Mariona Jové, Meritxell Martin-Garí, Manuel Portero-Otín

**Affiliations:** 1Ovivity Group, c/Angli 66 (torre), E08017 Barcelona, Spain; clarababot@ovivity.com; 2Metabolic Pathophysiology Research Group/Nutren-Nutrigenomics, University of Lleida-Lleida Biomedical Research Institute’s Dr. Pifarré Foundation (IRBLleida), Avda Rovira Roure, 80 E-25196 Lleida, Spain; ltsantos@ua.pt (L.S.); jceserrano@udl.cat (J.C.E.S.); mariona.jove@udl.cat (M.J.); meritxell.martin@udl.cat (M.M.-G.)

**Keywords:** fecundation, inflammation, cytokine, growth factors, metabolomics, lipidomics

## Abstract

Egg-yolk based supplements have demonstrated biological effects. We have developed a novel processed egg-yolk (PEY) complement, and we have tested whether it has inflammation modulatory properties. These were evaluated in a lipopolysaccharide (LPS)-challenge in 1-month male rats by in vivo circulating cytokine profiles measured by multiplexing techniques. Cell culture was used to explore ex vivo properties of derived serum samples. We explored growth factor composition, and mass-spectrometry metabolome and lipidome analyses of PEY to characterize it. PEY significantly prevented LPS-induced increase in IL-1 β, TNF-α, and MCP-1. Further, serum from PEY-treated animals abrogated LPS-induced iNOS build-up of the Raw 264.7 macrophage-like cell line. Immunochemical analyses demonstrated increased concentrations of insulin-like growth factor 1 (IGF-1), connective tissue growth factor (CTGF), and platelet-derived growth factor (PDGF) in the extract. PEY vs. egg-yolk comparative metabolomic analyses showed significative differences in the concentrations of at least 140 molecules, and in 357 in the lipidomic analyses, demonstrating the complexity of PEY. Globally, PEY acts as an orally-bioavailable immunomodulatory extract that may be of interest in those conditions associated with disarranged inflammation, such as inflammaging.

## 1. Introduction

Inflammaging is a chronic increase in the body’s pro-inflammatory status with advancing age in some tissues [1]. As inflammatory reactions increase, neurohormonal signaling (e.g., renin-angiotensin, adrenergic, insulin-IGF1 signaling) tends to be deregulated, immunosurveillance against pathogens and premalignant cells declines, and the composition of the peri- and extracellular environment changes, thereby affecting the mechanical and functional properties of tissues involved [2]. With the rise of this aging characteristic, a new concept of “anti-inflammaging” was also proposed, which influences progressive pathophysiological changes, as well as lifespan, and acts along with inflammaging [1]. In physiological conditions, the immune system helps to maintain homeostasis by mounting nonspecific innate and specific adaptive responses (inflammation) against potential aggressions. The inflammatory response is driven by a complex network of mediators and signaling pathways and is the net effect of interactions between pro-inflammatory and anti-inflammatory molecules (cytokines) that determines the immune response [3]. A properly functioning and reliable immune system is essential for maintaining health. When the expression of cytokines is chronically altered, it can lead to chronic inflammation, tumorigenesis, and autoimmunity. Inflammaging comprises the phenomena explaining the age-related trend towards increasing pro-inflammatory cytokine concentrations, more pronounced for IL-6, IL-8, IL-2, IFN-γ, and TNF-α [4], which is an indicator of the inflammation. It is logical to think, thus, that adequating the wrong cytokine profile, can prevent or slow down over some of the harmful consequences of the aging process.

Among the measures addressed to this rearrangement, pharmacological treatments can be found—statins to improve cardiac health [5]; also, nutritional approaches like caloric restriction [5]. Also, dietary supplements have emerged as potential anti-aging treatments, like antioxidants (curcumin, polyphenols), vitamins, and probiotics. The NIH defines dietary supplements as “substances you might use to add nutrients to your diet or to lower your risk of health problems, like osteoporosis or arthritis” [6]. Raising health concerns, changing lifestyles, and dietary habits, are driving increased attention to these products. In this line, there is ample scientific evidence that eggs contain biologically active compounds that may have a role in the therapy and prevention of chronic and infectious diseases. Several dietary supplements derived from eggs can be found in the market, broadly divided into unfertilized (commercialized), and fertilized. Commercialized hen eggs are highly nutritious food; its macronutrient content includes low carbohydrates and about 12 g per 100 g of protein and lipids (most of which are monounsaturated) as well as several nutrients such as zinc, selenium, retinol, and tocopherols. It is known that regular commercialized egg yolk contains proteins and peptides with biological activity. For example, it has been seen that whole egg yolk in the native form as a potential source of angiotensin converting enzyme-inhibitory peptides, or that yolk lipoproteins are important for lipid-mediated antimicrobial activity; also, a protein called γ-livetin, also referred to as immunoglobulin Y, exerts the same immunomodulatory activity as immunoglobulin G, and has been shown to have immunoregulatory effects [7]. In fact, in vivo anti-inflammatory and analgesic effects of nonfertilized egg yolk have been reported [8]. Based on preclinical studies, egg phosphatidylcholine and sphingomyelin species appear to regulate cholesterol absorption and inflammation; in clinical studies, egg phospholipid intake is associated with beneficial changes in biomarkers related to HDL reverse cholesterol transport [9]. Fertilized eggs contain every nutrient essential to sustaining a new life, in a closed shell that only permits O_2_, CO_2_ and H_2_O gas exchange [10], and its composition changes as the embryo develops. The screening of the bioactive components in the different development stages has attracted researcher’s attention, and important differences have been observed, showing very interesting potential applications. However, reports on the immune mechanism and the related bioactive components are limited.

Furthermore, to our knowledge, there are no studies regarding the effects of the intake of fertilized eggs *in vivo*. We have obtained, by a patented process [11], an extract of fertilized eggs, termed here patented egg yolk (PEY). To shed light over its effects in the inflammaging context, we explored its potential anti-inflammatory effect in comparison with the commercial egg yolk, and we characterized its composition in order to establish further screening protocols of the potential mechanisms of action, with the final goal of establishing the basis for potential interventional inflammaging studies.

## 2. Materials and Methods

### 2.1. Chemicals

Unless stated otherwise, all chemicals were obtained from indicated major commercial suppliers. PEY consists of an egg preparation obtained through a patented process [11]. It comprises a mixture of yolk and white extracted from a fertilized egg which has been incubated for a short period, which then is lyophilized to obtain a commercial product called Excelvit^®^.

### 2.2. Animals

One-month-old male Wistar rats (initial weight 75–100 g, *n* = 10 per group in the 5 day treatment regime, and *n* = 5 in the two or one day-treatment regime) obtained from Harlan Laboratories (Catalunya, Spain) were maintained at 23 ± 2 °C under a 12:12 h light-dark cycle (lights on from 07:00 to 19:00). All rats were allowed unlimited access to Harlan Teklad 14% protein rodent maintenance diet during the whole experimental period. After one week of acclimation to animal facilities, the animals were weighed and divided into two groups (PEY and egg yolk group) with equal bodyweight. Experimental treatment consisted of a 4-h fast (8:00–12:00 a.m.), where further a daily dose of 2000 mg/kg of PEY or egg yolk during one, two or five days was administered by oro-gastric gavage. Bodyweight and signs of toxicity were recorded daily during the treatment period, in which all treated animals showed adequate health. After the designated duration (one, two or five days), 2-h after the oro-gastric gavage of PEY or egg yolk, all animals were intraperitoneally injected with a 2.5 mg/kg dose of lipopolysaccharide (LPS, Sigma-Aldrich, Sant Louis, MO, USA). One hour after LPS injection, all animals were sacrificed by cervical displacement, and blood samples were collected by cardiac puncture. Serum from blood samples was collected within 30 min by centrifugation and immediately frozen with liquid nitrogen, and stored at −80 °C until further analyses. The study was conducted in accordance with the Declaration of Helsinki, and the protocol was approved by the University of Lleida Institutional Animal Care Committee (Project Code CEEA 03/03-12). 

### 2.3. Cytokine Analysis

Cytokines were determined by Milliplex RECYMAG65K27PMX magnetic bead immunology multiplex assay (Merck-Millipore, Burlington, MA, USA) following the manufacturer’s instructions. 

### 2.4. Cell Culture and Treatments

Mouse RAW 264.7 macrophages (ATCC TIB-71) were grown in a humidified incubator containing 5% CO_2_ and 95% air at 37 °C. Cells were grown in DMEM medium supplemented with 10% of heat-inactivated fetal bovine serum, 100 UI/mL of penicillin, and 100 µg/mL of streptomycin. In all experiments, cells were grown until confluence and then transferred to 6-well plates. On the day of the assay, the medium was changed to serum-free medium for 4 h, and further cells were exposed to DMEM medium supplemented with 10% rat serum obtained from rats exposed to a dose of 2000 mg/day (for five days) of PEY or egg yolk, or fetal bovine serum. For iNOS activation, a dose of 0.1 microgram/mL of LPS was added to selected wells. After 24-h, cells were collected and lysed using RIPA buffer with protease and phosphatase inhibitors. Protein concentrations were measured using the Bradford assay (BioRad Laboratories, München, Germany) with bovine serum albumin as a standard, and further cell-lysates were frozen a −80 °C for further analyses.

### 2.5. Western Blot Analysis

Total protein (15–40 μg) was resolved by SDS-PAGE and electroblotted onto polyvinylidene difluoride membranes (Immobilon-P Millipore, Bedford, MA, USA). Immunodetection was performed using primary antibodies anti-iNOS (Cell Signaling Technologies #2977), PDGF (RD Systems, AB23NA), NGF (Abcam Ab6199), CTGF (Genetex, GTX37727), and IGF-1 (Abcam, Ab106838). A monoclonal antibody to β-actin (Sigma, Saint Louis, MO, USA) was used to control protein loading from cell culture samples. Protein bands were visualized with the chemiluminescence ECL^®^ method (Millipore Corporation, Billerica, MA, USA). Luminescence was recorded and quantified in Lumi-Imager equipment (Boehringer, Mannheim, Germany), using the Quantity One 4.6.5. software (Bio-Rad, Hercules, CA, USA).

### 2.6. Lipidomic and Metabolomic Analyses

Metabolites and lipids in PEY and egg yolk were analyzed by liquid chromatography coupled to mass spectrometry as previously described [12]. For metabolomics, samples were depleted of proteins by methanol addition [13]. The resulting metabolites were separated using a reverse-phase column (Zorbax SB-Aq 1.8 µm 2.1 × 50 mm; Agilent Technologies, Barcelona, Spain). We employed a gradient of water to methanol (all with 0.2% acetic acid) in a chromatograph (LC Agilent 1290) coupled to a mass spectrometry system (time of flight (TOF) mass spectrometer Agilent 6520). Mass spectra were collected in both negative and positive ionization modes, scanning from *m*/*z* values from 50 to 1600, at 1.5 scans/s. For lipidomic analyses, lipids were obtained by chloroform:methanol extraction, internal standards added, and processed as described [14,15] employing the same system as above (Agilent Technologies, Santa Clara, CA, USA) with adequate solvents [14]. This method allows the orthogonal characterization (based on exact mass (<10 ppm) and on retention time) of lipids. When combined with internal standards, this strategy is useful for proposing potential identities with low uncertainty [14,16]. In this case, data were collected in both positive and negative electrospray ionization in full-scan mode at 100–3000 *m*/*z* in an extended dynamic range (2 GHz). 

### 2.7. Statistical Analyses and Data Annotation

Statistic calculations were performed using SPSS (IBM SPSS v 25, Armonk, NY, USA) and GraphPad Prism 8 (GraphPad Software, San Diego, CA, USA). The Kolmogorov–Smirnov test checked the normality of the distribution of variables. For metabolomics and lipidomic analyses, only common features (found in ≥75% of the replicas of PEY and egg yolk samples) were taken into account, to correct for individual bias. Principal component analysis (PCA), partial least squares discrimination analysis (PLS-DA), and hierarchical clustering analysis were performed using MassHunter Mass Profiler Professional (Agilent Technologies, Barcelona, Spain) after the transformation of chromatographic results to the CEF^®^ format. Analyses (Volcano plots, PCA and PLS-DA reported here) were performed employing the Metaboanalyst 3.0 platform [17].

Annotations of metabolites were produced based on an accurate mass-retention time algorithm (Agilent^®^, MassHunter Mass Profiler Professional) and for lipids on the comparison with the retention time of internal standards. Therefore, the level of evidence of annotation is 2 (i.e., putatively annotated compounds, without chemical reference standards, based upon physicochemical properties and spectral similarity with public/commercial spectral libraries), according to [18]. A level of *p* < 0.05 was selected as the point of minimal statistical significance in every comparison. Pathway analyses were produced according to these putatively annotated compounds by employing the ConsensusPathDataBase platform [19]. 

## 3. Results

As expected, LPS injection induced an increase in plasma levels of cytokines (Appendix A), increasing ca 43% circulating levels of TNF-α after 2 h. To study the systemic anti-inflammatory effects of PEY, plasmatic concentrations of cytokines of animals feed during five days with PEY were measured after 2 h of performing the inflammatory stimulus (LPS injection) (Figure 1A). Compared with egg-yolk, the PEY group presented a significant reduction in plasmatic concentrations of IL-1 β (Figure 1B), TNF-α (Figure 1C), and MCP-1 (Figure 1D). Interestingly, shorter times of treatment also sufficed to abrogate LPS-induced buildup of circulating cytokines (1 day, Appendix A; or 2 days, Appendix A).

Demonstrating the bioavailability of PEY and the solubility of PEY-induced effects, the anti-inflammatory effect was confirmed ex-vivo using Raw 264.7 cells treated with the serum of the animals exposed during five days to PEY and egg-yolk (Figure 1E). RAW 264.7 cells were treated with LPS to stimulate their inflammatory response, and i-NOS content was measured. A significant difference in the iNOS immunoreactivity between the LPS group and the other groups, showing the different inflammatory responses of the cells, was observed (Figure 1E). iNOS production by the cells treated with the serum of the egg yolk group was significantly lower than control, and the serum of rats treated with PEY had the highest anti-inflammatory effect, significantly lower than the egg yolk.

The composition of PEY was studied to explore the origin of the anti-inflammatory effects observed. We evaluated the presence of growth factors (Figure 2), as well as the comparative metabolomics and lipidomics (Figure 3) of egg yolk and PEY.

It was observed that PEY contained 2.5 more connective tissue growth factor (CTGF), 1.7 more platelet-derived growth factor (PDGF), 1.3 more nerve growth factor (NGF) and 8.6 more insulin-like growth factor (IGF-1) than egg yolk (see Figure 2). All the increases were significant, but NGF. The presence of growth factors has been previously described [20]. The results of liquid chromatography coupled to mass spectrometry indicate (Figure 3) that PEY contains a non-negligible number of methanol-soluble (metabolites) and organic solvent-soluble (lipids) differing from egg yolk. Interestingly, lipidomic profiles allow a more thorough differentiation in comparison to metabolomics (Figure 3A,E). Even applying a Benjamini–Hochberg correction for false discovery rate, a total of 357 differential lipids were found (231 increased in PEY and 126 increased in PEY, FDR = 0.05; Supplemental Dataset). A total number of 140 metabolites differentially present in the extract were also found (16 increased in egg yolk and 124 increased in PEY, FDR = 0.05; Supplemental Dataset). Collectively, regarding putative identifications, these lipids clustered among different sphingolipid-related pathways, as well as some immune-related pathways (Figure 3D). Interestingly, differential metabolites were also present in pathways related to omega-3 fatty acids, among many other metabolic nodes (Figure 3H). All differential metabolites and lipids, including exact mass, chromatographic behavior (retention time in respective systems), composite spectrum, fold-change and *p*-values (both raw and adjusted for false-discovery rate) are available as a Supplemental Dataset in Excel^®^ file in Appendix A, found online. Similarly, the individual levels of metabolites and lipids employed for pathway analyses (Figure 3D,H) can be found in an ad-hoc designed web page (https://excelv.herokuapp.com/).

## 4. Discussion

It is known that nutrition can affect the functioning of various immune parameters, and the immunomodulation through dietary supplements is not new; e.g., linoleic acid promotes the production of leukotrienes and prostaglandins, or that arginine and glutamine enhance macrophage phagocytosis [21]. Also, epidemiological studies show that both overall diet or specific dietary components like polyphenols can reduce inflammatory cytokines in animal and cell culture models [22]. Then, it was expectable that a dietary component so rich in biomolecules as egg yolk would lead to some effects in the immunity. What it was not predictable, is the fact that PEY would significantly improve these effects.

In our view, PEY optimizes the immune response. This effect means that, more than down-regulating hyperactive immune functions or up-regulating suppressive immune functions, PEY strengthens the resilience of immune functions to respond to external ‘stressors’ [21]. The results of this work agree with PEY intake induced changes at the systemic level that led to a less aggressive immune response in rodents since a lower induction of pro-inflammatory cytokines was observed after an inflammatory stimulus. PEY, like any biological extract, is a complex matrix containing thousands of molecules with biological activity, in a vast range of concentrations; thus, elucidating its mechanism of action is not an easy task. The individual effects of single compounds can be easily studied in vitro. However, synergisms and antagonisms, as well as metabolic reactions, take place when a food, as a complex matrix, is ingested, digested, and processed by a living being. Accounting for cytokine blood levels, it was clear that some differences in the composition between PEY and egg yolk must exist, and screening these differences could give some clues. 

The increased presence of growth factors (CTGF, PDGF, IGF-I and NGF) on PEY, when compared with egg yolk, is not surprising from a biological point of view. Growth factors play a vital role in the evolution and resolution of inflammatory reactions [22]. Thus the presence of these proteins leads to thinking that they could be related to the different immune responses observed in vivo. CTGF is a critical player in connective tissue homeostasis since it helps to maintain extracellular matrix remodeling in normal physiological processes such as wound healing, and it has also been shown to possess apoptotic and nonmitogenic properties [23]. PDGF has a vital role in the early differentiation of hematopoietic/endothelial precursors; it has been used in clinical trials as a topical treatment for healing chronic neuropathy, as well as to improve periodontal regeneration in severe periodontal disease [24]. Insulin-like growth factor I (IGF-I) is a polypeptide hormone secreted by multiple tissues in response to growth hormone (GH). It is partly responsible for GH activity, and also has anabolic effects. NGF is a neurotrophic factor that promotes the growth and survival of peripheral sensory and sympathetic nerve cells. It is a pleiotropic factor, since it produced and utilized by several cell types, including structural (epithelial cells, fibroblasts/myofibroblasts, endothelial cells, smooth muscle cells and hepatocytes), accessory (glial cells, astrocytes and Muller cells) and immune (antigen presenting cells, lymphocytes, granulocytes, mast cells and eosinophils) cells [25], having neuroprotective and tissue repairing properties. Strikingly, since PEY is administered orally, it must be assumed that there exists bioavailability of these molecules. PEY effects could be explained through absorption in the digestive tract by indirect interaction with the host’s microbiota. Previous studies showed how the Platelet-Rich Plasma rich in growth factors (PDGF, IGF, VEGF, TGFb) promoted the regenerative processes inhibiting the macrophage activation and the release of cytokines (TNFa, MCP-1, and RANTES) [26] in vitro. However, another study showed that orally administered IGF-I mainly acts at the intestine, a portion of ingested IGF-I is absorbed into the general circulation to enhance the growth of selective organs/tissue [27]. As indicated, interaction with microbiota cannot be excluded—for instance, Padlyia et al. [28] analyzed fertilized eggs, finding that the proteins that increased in abundance play a role in angiogenesis (pleiotrophin, histidine-rich glycoprotein), in defending the developing embryo against microbial pathogens (avian β-defensin 11, polymeric immunoglobulin receptor, serum amyloid P-component, ovostatin and mannose-binding ligand) and in augmenting the structural integrity of the egg shell (ovo-calyxin-32), necessary to provide a substantial barrier against microbial infection.

The action of PEY is not only due to the presence of growth factors. The lipids and metabolites that were found increased in PEY clustered among different sphingolipid-related pathways, as well as some immune-related pathways. A plethora of cell biological processes are critically modulated by bioactive sphingolipids, including growth regulation, cell migration, adhesion, apoptosis, senescence, and inflammatory responses [29]. Similarly, Duan et al. [30] reported that fertilized eggs exhibited higher essential fatty acids (EFAA) and monounsaturated (MUFA) fatty acids levels than unfertilized eggs, and lower cholesterol concentrations, having the potential of being utilized as an EFAA/MUFA-rich, low-cholesterol dietary supplement for the aged and people with special dietary requirements. There are also in vitro experiments demonstrating the pharmacological effects of these bioactive molecules; Xi Li et al. [31] observed that 12-day chicken embryo extracts enhanced spleen lymphocyte proliferation (and IL-2 production), and macrophage function (phagocytosis and NO production). Accounting the high bioavailability of lipids, we cannot discard a role of lipids in PEY in the immune optimization.

Globally, the observed decrease in LPS-induced increase in circulating TNFa, IL1b, and MCP-1 could be related with the complex PEY composition. Growth factors, together with a vibrant matrix of bioactive lipids and metabolites, are promoting a less aggressive immune response, which means better maintenance of homeostasis. Further, these responses can be ascribed to a change in the M1-M2 macrophage status, favoring the resolutive (M2) phase, a thread that will be the focus of future studies.

As for the limitations of our work, we acknowledge that a single nutrient cannot explain observed effects; that the oral bioavailability of the compounds present in PEY can be limited; and that perhaps the system is limited to preclinical studies. Further, we think that measurement of systemic cytokine data induced by PEY treatment, as well as its surrogate metabolome and lipidome changes, the focus of future studies, could offer more light on the potential mechanisms behind this. However, we think that the demonstration that PEY is a complex of bioactive molecules (growth factors, lipids, metabolites) that encloses the molecules that an embryo needs to develop a whole organism affects the host’s immunity is clear. It is reasonable to think that these active compounds are bioavailable even if orally administered since a systemic effect has been observed. These active molecules probably act together; this differentiates it from a single active ingredient since possibly this set of molecules produces a series of simultaneous effects that include synergies and antagonisms, promoting an anti-inflammatory microenvironment and leading to a state of homeostasis. Since chronic inflammation contributes to the development of chronic diseases (cancer, cardiovascular disease, and diabetes) and aging, consumption of PEY could effectively reduce the incidence and the progression of these processes.

## Figures and Tables

**Figure 1 nutrients-12-02699-f001:**
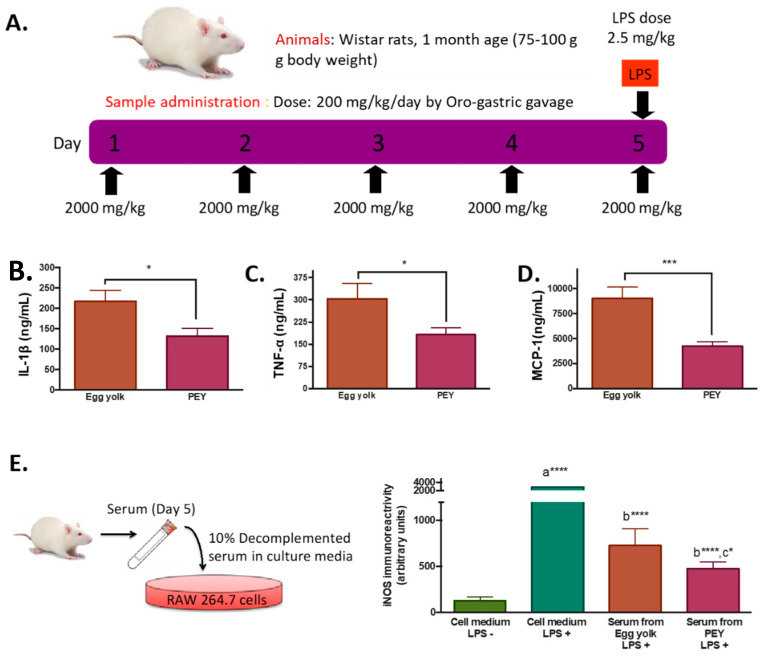
Characterization of the anti-inflammatory effects of processed egg-yolk in vivo. (**A**). Oral administration of processed egg-yolk (PEY) during five days and application of inflammatory aggression the 5th day by lipopolysaccharide (LPS). (**B**–**D**). Plasmatic concentrations of cytokines IL-1β, TNF-α, and MCP-1 in the control group (egg yolk) and treated group (PEY) after 2 h of the administration of lipopolysaccharide (LPS) (* *p* < 0.05, *** *p* < 0.001). (**E**). iNOS immunoreactivity of Raw 264.7 cell culture exposed to 0.1 µg/mL of LPS after the application of standard cell medium (no treatment), blood serum from rats treated with egg yolk (control) and serum from rats treated with PEY; a **** significant difference with cell medium LPS− *p* < 0.001; b **** significant difference with cell medium LPS+ *p* < 0.0001; c * significant difference with serum from egg yolk LPS+ *p* < 0.05. One-way ANOVA or Student’s *t*-test was used for statistical analyses. In vitro experiments were performed in triplicate, while as treatments were delivered to *n* = 10 animals per group.

**Figure 2 nutrients-12-02699-f002:**
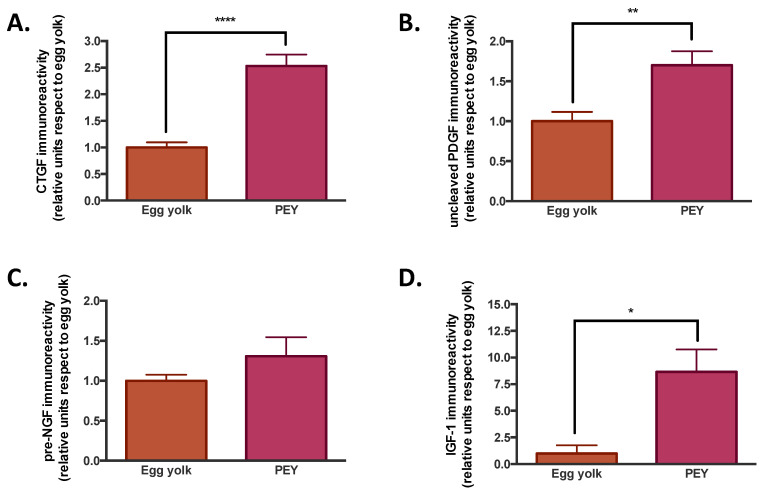
PEY contains a significant amount of growth factors. (**A**). Connective tissue growth factor (CTGF) immunoreactivity in egg yolk and PEY, showing a difference between them with a *p* < 0.0001 significance. (**B**). Platelet-derived growth factor (PDGF) immunoreactivity in egg yolk and PEY, showing the difference between them with a *p* < 0.01 significance. (**C**). Nerve growth factor (NGF) immunoreactivity in egg yolk and PEY, showing no significant difference between them. (**D**). Insulin-like growth factor 1 (IGF-1) immunoreactivity in egg yolk and PEY, showing a significant difference between them (* *p* < 0.05, ** *p* < 0.01, **** *p* < 0.0001).

**Figure 3 nutrients-12-02699-f003:**
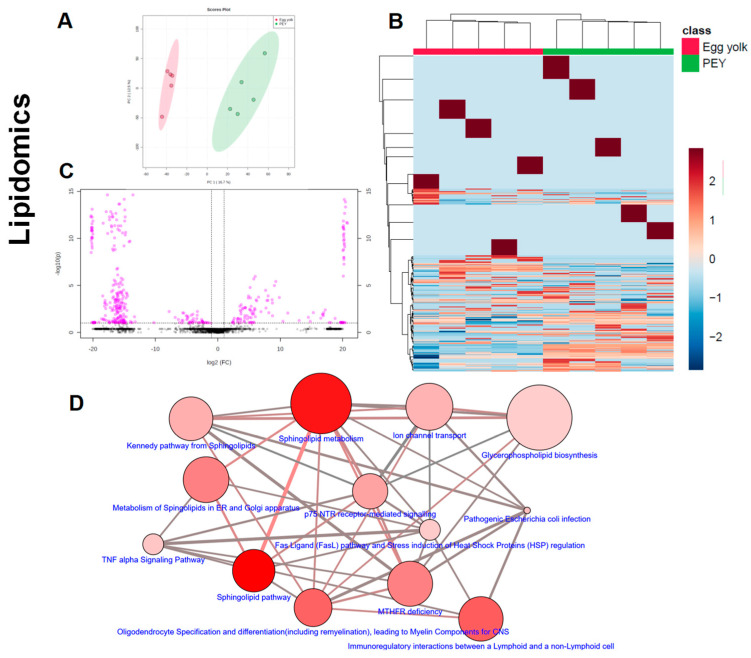
PEY exhibits a differential lipidomic and metabolomic profile. Lipidome (**A**–**D**) and metabolome (**F**–**H**) profiles of PEY and egg yolk were analyzed by liquid chromatography coupled to mass spectrometry. As shown by principal component analyses graphs, PEY and egg yolk show different lipidome (**A**) and metabolome (**E**) signatures. This is reinforced by hierarchical clustering analyses which show a perfect clustering in lipidomics (**B**), but not in metabolomics (**F**). Univariate statistics such as a Volcano plot are shown, indicating that there are marked differences in a high number of molecules both in the lipidome (**C**) and metabolome (**G**). Pathway analyses, shown in (**D**) for lipidomics and in H for metabolomics, indicate the pathways where putatively annotated molecules are located—network neighborhood-based entity sets of lipidomic and metabolomics differences between PEY and egg yolk. Differential lipids (**D**) and metabolites (**H**) (see main text) were entered into the ConsensusPathDB platform, and nodes, representing neighborhood-based entity sets (whose size is proportional to the number of metabolites/lipids of the set, and color intensity denote *p*-value for hypergeometric tests) are linked by interactions consisting of the number of metabolites shared by nodes. The type of network chosen was 2-next neighbors, with a minimum number of 1 metabolite overlap with members of the entity set (and a *p* < 0.05 for lipidomics and a *p* < 0.01 for metabolomics as the cutoff). The sets were obtained, considering only Wikipathway based ones. Lipids included were C21480, C21481, C00350, C02737, C13883, C00195, C12126, C00550, C01190, and C02686. Metabolites used were C01179, C06104, C06425, C00239, C00364, C00984, C14214, C02043, C06429, C08491, C00256, C05332, C00410, C05441, and C02477 (KEGG nomenclature). Individual levels of metabolites and lipids employed for pathway analyses are available in https://excelv.herokuapp.com/.

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
