# Peer review of "In Vivo Anti-Inflammatory Effects and Related Mechanisms of Processed Egg Yolk, a Potential Anti-Inflammaging Dietary Supplement"

_nutrients, 2020, doi:10.3390/nu12092699_

Round 1

Reviewer 1 Report

In this paper, the authors propose that an egg-yolk-based commercial product (called Excelvit) does exhibit significant antiinflammatory properties, doing even better than regular natural hen egg yolk. some of the properties already observed for some hen egg components (e.g. some hen egg proteins and peptides do exert ACE-inhibitory effects) can be found in the proposed mixture. In addition, the commercial mix seems more performant than the original hen egg yolk. Both lipidomics and proteomics seem to support the findings.

Author Response

In this paper, the authors propose that an egg-yolk-based commercial product (called Excelvit) does exhibit significant antiinflammatory properties, doing even better than regular natural hen egg yolk. some of the properties already observed for some hen egg components (e.g. some hen egg proteins and peptides do exert ACE-inhibitory effects) can be found in the proposed mixture. In addition, the commercial mix seems more performant than the original hen egg yolk. Both lipidomics and proteomics seem to support the findings.

We thank the reviewer for the kind appreciation given to our work.

Reviewer 2 Report

The manuscript by Cunill et al., aims to understand the potential inflammation modulatory properties of processed egg yolk (PEY). The authors provide some evidence using both in vivo and ex vivo approaches to support their claim that PEY possesses anti-inflammatory properties. Collectively, these studies appear well performed and suggest that PEY modulates inflammatory responses beyond those of egg yolk alone. Nevertheless, there are some issues that deserve attention.

Major:

        Cytokine data presented in Figure 1A-D compare the plasma levels of pro-inflammatory cytokines in PEY and egg yolk treated animals following LPS stimulation. The data suggest that PEY treatment reduces the cytokine response to LPS administration however the degree of such reduction is difficult to infer without proper controls. Authors should include data from vehicle treated animals so that readers can ascertain the degree of change with PEY treatment.

         Data presented in Figures 2 and 3 are component characterizations of growth hormones, lipids, and metabolic profile between egg yolk and PEY extracts. More useful would be to include systemic cytokine data along with data on plasmatic changes in animals following egg yolk and PEY supplementation as data in RAW macrophages alone (Fig 1E) are not enough to determine possible sources of systemic cytokine production. 

        Authors should consider presenting data for robust changes in lipids and metabolic profiles from Fig 3. Pathway analysis is helpful however changes in specific lipids that are the major drivers in group differences are not presented. 

      More details regarding methodological approaches including "n" values are recommended to enhance confidence of both in vivo and ex vivo data.

Author Response

The manuscript by Cunill et al., aims to understand the potential inflammation modulatory properties of processed egg yolk (PEY). The authors provide some evidence using both in vivo and ex vivo approaches to support their claim that PEY possesses anti-inflammatory properties. Collectively, these studies appear well performed and suggest that PEY modulates inflammatory responses beyond those of egg yolk alone. Nevertheless, there are some issues that deserve attention.

We thank the reviewer for the careful reading and editorial work performed. We have introduced suggested changes, as detailed below

Major:

        Cytokine data presented in Figure 1A-D compare the plasma levels of pro-inflammatory cytokines in PEY and egg yolk treated animals following LPS stimulation. The data suggest that PEY treatment reduces the cytokine response to LPS administration however the degree of such reduction is difficult to infer without proper controls. Authors should include data from vehicle treated animals so that readers can ascertain the degree of change with PEY treatment.

We agree with the reviewer. Our control, in the experiments presented in Figure 1A-D, were considered animals fed with the egg yolk, as this is the nutritional ingredient related to the PEY. In order to understand the important question raised by the reviewer, we have included in the revised version, results from animals treated with vehicle (water), as well as the effects from the LPS untreated animals.

Thus, in the Supplemental Figure 1, now it appears this information.

SUPPLEMENTAL FIGURE 1

Supplemental Figure 1. A). Oral administration of PEY during one day is able to prevent LPS-induced TNF-α increase in serum in comparison with water or egg yolk. B) Oral administration during two days is able to prevent LPS-induced increase of serum cytokines IL-1β, TNF-α, IFN-γ, and IL-2.  *** indicates significant differences between groups (p<0.001) in two- way ANOVA post-hoc analyses after Benjamini, Krieger and Yekutelli correction for false discovery rate. In all cases, 5 animals were treated per group.

We have also ammended the results section to show these vehicle (water)-treated animals.

“As expected, LPS induced an increase in plasma levels of cytokines (Supplemental Figure 1A), increasing ca 43% circulating levels of TNF-α. To study the systemic anti-inflammatory effects of PEY, plasmatic concentrations of cytokines of animals feed during five days with PEY were measured after 2 hours of performing an inflammatory stimulus (LPS injection) (Figure 1A). Compared with egg-yolk, PEY group presented a significant reduction in plasmatic concentrations of IL-1 β (Figure 1B), TNF-α (Figure 1C), and MCP-1 (Figure 1D). Interestingly, shorter times of treatment also sufficed to abrogate LPS-induced buildup of circulating cytokines (1 day, Supplemental Figure 1A; or 2 days, Supplemental Figure 1B).  “

         Data presented in Figures 2 and 3 are component characterizations of growth hormones, lipids, and metabolic profile between egg yolk and PEY extracts. More useful would be to include systemic cytokine data along with data on plasmatic changes in animals following egg yolk and PEY supplementation as data in RAW macrophages alone (Fig 1E) are not enough to determine possible sources of systemic cytokine production. 

We thank that interesting observation. These measurements would be focus of future studies. However, the observations presented in supplemental figure 1A, where we measured TNFlevels in LPS untreated animals, reveal no basal changes in this scenario. Further, we have included this issue in the limitations section of the revised version:

“As for the limitations of our work, we acknowledge that a single nutrient cannot explain observed effects; that the oral bioavailability of the compounds present in PEY can be limited; and that perhaps the system is limited to preclinical studies. Further, we think that measurement of systemic cytokine data induced by PEY treatment, as well as its surrogate metabolome and lipidome changes, focus of future studies, could offer more light on the potential mechanisms behind.”

        Authors should consider presenting data for robust changes in lipids and metabolic profiles from Fig 3. Pathway analysis is helpful however changes in specific lipids that are the major drivers in group differences are not presented. 

We thank the reviewer for this. In the revised version, we have prepared an interactive web page showing the specific lipids and metabolites contributing to group differences. This web page is found in https://excelv.herokuapp.com/. Specific examples of this web page appear below

      More details regarding methodological approaches including "n" values are recommended to enhance confidence of both in vivo and ex vivo data.

We thank the reviewer for this indication. As shown in the revised version of the manuscript, we have included all the n values in the different experiments. For instance, in the revised Figure 1 legend, now it appears:

“Figure 1. A). Oral administration of PEY during five days and application of inflammatory aggression the 5th day by lipopolysaccharide (LPS). B,C,D). Plasmatic concentrations of cytokines IL-1β, TNF-α, and MCP-1 in the control group (egg yolk) and treated group (PEY) after 2 hours of the administration of LPS. E). iNOS immunoreactivity of Raw 264.7 cell culture exposed to 0.1 µg/mL of LPS after the application of standard cell medium (no treatment), blood serum from rats treated with egg yolk (control) and serum from rats treated with PEY; a**** significant difference with cell medium LPS- p<0.001; b**** significant difference with cell medium LPS+ p<0.0001; c* significant difference with serum from egg yolk LPS+ p<0.05. One-way ANOVA or Student’s t-test were used for statistical analyses. In vitro experiments were performed in triplicate, while as treatments were delivered to n=10 animals per group.”

Round 2

Reviewer 2 Report

The reviewer appreciates the author's response to recommendations. Including most prominent and significant findings from the lipidomic and metabolomic datasets into the supplemental material as well as the online website is recommended. 

Author Response

The reviewer appreciates the author's response to recommendations. Including most prominent and significant findings from the lipidomic and metabolomic datasets into the supplemental material as well as the online website is recommended. 

We thank the reviewer for kind appreciation given to the paper and careful feedback, that has improved our paper’s quality. Regarding this last request, in the revised version, we have now included this sentence, in the Results section

. All differential metabolites and lipids, including exact mass, chromatographic behavior (retention time in respective systems), composite spectrum, fold-change and p values (both raw and adjusted for false-discovery rate) are available as a Supplemental Dataset in Excel® file in Supplemental materials, found online. Similarly, the individual levels of metabolites and lipids employed for pathway analyses (Figures 3D and 3H) can be found in an ad-hoc designed web page ( https://excelv.herokuapp.com/)”